# Dermatofibrosarcoma Protuberans of the Vulva: A Review of the MITO Rare Cancer Group

**DOI:** 10.3390/cancers16010222

**Published:** 2024-01-03

**Authors:** Rosanna Mancari, Raffaella Cioffi, Francescapaola Magazzino, Laura Attademo, Miriam Sant’angelo, Gianluca Taccagni, Giorgia Mangili, Sandro Pignata, Alice Bergamini

**Affiliations:** 1Gynecologic Oncology Unit, IRCCS Regina Elena National Cancer Institute, 00144 Rome, Italy; 2Department of Obstetrics and Gynaecology, San Raffaele Scientific Institute, 20132 Milan, Italy; cioffi.raffaella@hsr.it (R.C.); mangili.giorgia@hsr.it (G.M.); bergamini.alice@hsr.it (A.B.); 3Complex Operating Unit Ginecologia E Ostetricia, Ospedale Civile Di San Dona’ Di Piave (Venezia), Aulss4 Veneto Orientale, 30027 San Donà di Piave, Italy; framagazzino@hotmail.it; 4Oncology Unit, Ospedale del Mare, 80147 Naples, Italy; laura.attademo@gmail.com; 5Department of Surgical Pathology, San Raffaele Scientific Institute, 20132 Milan, Italy; santangelo.miriam@hsr.it (M.S.); taccagni.gianluca@hsr.it (G.T.); 6Department of Urology and Gynecology, Istituto Nazionale Tumori IRCCS ‘Fondazione G Pascale’, 80144 Napoli, Italy; s.pignata@istitutotumori.na.it; 7Faculty of Medicine, Vita-Salute San Raffaele University, 20132 Milan, Italy

**Keywords:** vulvar dermatofibrosarcoma, dermatofibrosarcoma protuberans, vulvar cancers, rare gynaecological tumours

## Abstract

**Simple Summary:**

Rare diseases represent a major health problem, since patients face difficulties in obtaining a rapid diagnosis and appropriate treatments. Vulvar dermatofibrosarcoma protuberans is one of these rare entities, in which reaching a correct pathological diagnosis is intricate and surgical techniques are not standardised. The aim of our paper is to review the available literature on vulvar dermatofibrosarcoma protuberans to summarise previous experiences and main issues, in an attempt to improve the management of this rare disease. Dermatofibrosarcoma protuberans of the vulva needs to be diagnosed early and managed by a referral centre, where the patient can receive appropriate management: surgical treatment should aim to obtain free margins, lowering the probability of recurrence. Long-term follow up is needed, since recurrences are documented even after several years.

**Abstract:**

Background: Vulvar dermatofibrosarcoma protuberans is an extremely rare disease. Its rarity can hamper the quality of treatment; deeper knowledge is necessary to plan appropriate management. The purpose of this review is to analyse the data reported in the literature to obtain evidence regarding appropriate disease management. Methods: We made a systematic search of the literature, including the terms “dermatofibrosarcoma protuberans”, “vulva”, and “vulvar”, alone or in combination. We selected articles published in English from two electronic databases, PubMed and MEDLINE, and we analysed their reference lists to include other potentially relevant studies. Results: We selected 39 articles, with a total of 68 cases reported; they were retrospective case reports and case series. Dermatofibrosarcoma protuberans of the vulva tends towards local recurrence; an early and timely pathological diagnosis, together with an appropriate surgical approach, are of utmost importance to ensure free margins and maximise the curative potential. Conclusions: Even if this is an indolent disease and it generally shows a good prognosis, appropriate management may help in reducing the rate of local recurrences that may hamper patients’ quality of life. Management by a multidisciplinary team is highly recommended.

## 1. Introduction

Dermatofibrosarcoma protuberans (DFSP) is a rare, slow-growing, well-differentiated mesenchymal tumour arising in the dermis and usually with extension into subcutaneous tissue [1].

DFSP can occur anywhere, but the preferred sites are the trunk and extremities; a vulvar location is extremely rare, with fewer than 70 cases reported in the literature. 

First described by Darier and Ferrand in 1924 [2], vulvar DFSP is characterised by slow growth, rarely leading to distant spread (less than 5% of cases); local recurrence is common, ranging between 20 and 50% of cases. Due to its indolent course, diagnosis is often made when the disease is locally advanced. Moreover, reaching a correct histopathological diagnosis is challenging, as vulvar DFSP is often misdiagnosed with other more common tumours.

The gold standard for both first diagnosis and recurrence is represented by radical surgery, aiming at complete excision with free surgical margins. 

Given its rarity and the lack of available clinical guidelines for its management, treating these patients poses several challenges, from diagnosis to follow up. 

The aim of the present review is to systematically collect and analyse all of the available literature, and then summarise and discuss the evidence on vulvar DFSP.

## 2. Materials and Methods

A systematic search of the literature, until February 2023, was performed in two electronic databases (PubMed, MEDLINE and Embase) in order to identify articles relevant to the purpose of this systematic review. The article research was carried out according to the PRISMA (Preferred Reporting Items for Systematic Reviews and Meta-Analyses) framework [3], as presented in Figure 1. The search included the following keywords and medical subject heading terms, alone or in combination: “dermatofibrosarcoma protuberans”, “vulva”, and “vulvar”. All identified articles were examined and their reference lists were reviewed in order to include other potentially relevant studies. Two independent authors reviewed the studies (RM, AB) for inclusion. Discordant cases were discussed with a third author (FM). Eligibility for inclusion was initially assessed on the basis of titles and abstracts. The decision for final inclusion was made after the detailed examination of the full manuscripts.

The inclusion criteria were articles (also considering case series and case reports on vulvar DFSP) published in English in peer-reviewed journals between 1976 and 2023. Reviews or articles including tumours with mixed/other histologies or DFSP outside of the vulva were excluded.

## 3. Results

### 3.1. Study and Patient Characteristics

Figure 1 illustrates the flow of the systematic literature review. In total, 39 articles were selected for inclusion in this review, with a total of 68 cases reported (listed in Table 1). None of these were prospective series, while *n* = 32 were retrospective case reports and *n* = 7 case series. The number of patients included for each report ranged from 1 to 13. 

This review also includes one representative case from a MITO centre that has not been previously published.

Sixty-nine cases of vulvar DFSP meeting our inclusion criteria were reported, with a median age of 46 (range: 19–83). The most common site of presentation was the labia majora (52.2%), followed by the mons pubis (11.6%). The mean size of the lesion at the time of surgery was 5.32 cm (data available for 61 cases, range 1.0–20.0 cm). Vulvar DFSP is usually described as an asymptomatic vulvar subcutaneous and firm mass, and less commonly as a plaque-like lesion. In rare cases (7.2%), the presence of a vulvar mass has been associated with pain, itching, malaise, bleeding, and dyspareunia. 

The median time between the detection of the vulvar lesion and treatment was 24 months (range 1–252, data available for 29 patients, Table 1). In the included articles, the initial diagnosis was inconsistent with the final review in 22% of cases.

### 3.2. Pathology

Macroscopically, DFSP presented as a plaque-like cutaneous lesion, flat or elevated, firm, with irregular borders and of variable size. At the cut surface, it appeared as a single or multinodular lesion, with a translucent and gelatinous appearance, involving dermis and spreading into subcutaneous tissue.

Microscopically, the majority of our cases had the typical aspect of DFSP, presenting as low to intermediate differentiated tumours composed of spindle cells embedded in a collagenous stroma; in 3/69 cases (4.3%), the stroma was described as myxoid. In DFSP, tumour cells are typically arranged in a storiform pattern and show the entrapment of subcutaneous adipose tissue with a sparing of adnexal structures (“honeycomb” pattern). The cytoplasm is scant, eosinophilic, and fibrillary; the nuclei have low-grade atypia and low mitotic activity (Figure 2 and Figure 3). The presence of higher nuclear pleomorphism and increased mitotic count indicates the presence of fibrosarcomatous transformation (DFSP-FS) and was reported in 9/69 cases (13%) [11].

Immunohistochemical staining for CD34 was performed in 71% of patients (49/69), showing diffuse and strong expression in all cases of DFSP, except for those with fibrosarcomatous transformation, where staining was low or negative [11,15,28]. Vimentin was always positive, while staining for Desmin was negative in all available cases: S-100 was negative in 90% of cases.

Molecular studies have described that DFSP may often harbour a common chromosomal translocation t (17;22) (q22;q13) with the COL1A1-PDGFB fusion gene between the collagen type Iα1 gene (COL1A1) and the platelet-derived growth factor β-chain gene (PDGFB). The analysis of this rearrangement has only been recently performed; for this reason, this information is available in our records for 39/69 patients with vulvar DFSP (42%), with a positivity of 75.8%.

### 3.3. Treatment and Clinical Course

The details regarding treatment and clinical course are summarised in Table 2. Surgical excision with tumour-free margins is the gold standard of treatment for this disease. For limited volume lesions, wide local excision (WLE) was the most commonly applied surgical technique at primary surgery (61/68 = 89.7%); in cases of positive margins (26/65 = 40%), repeated surgery with WLE or vulvectomy has usually been proposed. 

Mohs microsurgery (MMS) was successfully applied in two cases following the positive experience of DFSP affecting other disease sites. These previous experiences have reported a lower rate of recurrence with this technique compared to wide excision (1.6% vs. 20%) [42].

Lymph node involvement has never been detected, confirming that lymphadenectomy is not recommended.

Adjuvant therapy is generally not recommended when radical excision is feasible. According to the present literature review, medical treatment with Imatinib was only offered in three cases, as neoadjuvant treatment for a large unresectable lesion or as adjuvant treatment in case of incomplete resection [11,15,17]. Adjuvant RT was administered in four cases: in two of them for positive margins after excision, in one case for local recurrence after WLE, and in one case for local recurrence without local excision [11,26,28,30].

Relapses of DFSP were frequent (20/68 = 29.4%). Most of the recurrences occurred locally (19/20 = 95.0%), particularly in cases of positive margins at local excision. In this analysis, the local recurrence rate was 42% in the case of positive margins (11/26) vs. 10.8% in the case of negative margins (4/37) (*p* = 0.003).

Distant spread was rare (3/20 cases = 15%), with the most commonly involved site being the lung; one of these cases was a DFSP-FS, and in the remaining two cases, classic DFSP was diagnosed. Notably, one of the patients experiencing relapse did not attend the recommended follow-up schedule [13,22,30]. Lung metastases were treated with Imatinib in one case, achieving partial response [22], or with conventional chemotherapy [30]. 

Deaths from disease have been rarely reported (2/68 = 2.9%), and in both cases, they were related to the presence of distant metastases. 

## 4. Discussion

The present analysis confirms that vulvar DFSP is an indolent disease, with slow growth and a tendency towards local recurrence. Despite its general good prognosis, this disease requires proper management, given the high incidence of local and repeated recurrences that may negatively impact quality of life.

Given the rarity of this disease, many challenges in diagnosis and management need to be faced, from the correct and timely diagnosis to radical surgery. For this reason, management by an expert multidisciplinary team is highly recommended.

Early diagnosis is of utmost importance to allow appropriate and conservative surgery; since DFSP is usually paucisymptomatic at first presentation, diagnosis is often made several months after tumour appearance. Diagnostic delay is also conditioned by DFSP being an extremely rare tumour that uncommonly presents in the vulva; thus, achieving a correct final diagnosis is challenging. 

The spindled cells are usually arranged in a storiform pattern and are typically associated with minimal cytologic atypia. Immunohistochemistry for CD34 is mostly positive. The presence of DFSP-FS is associated with a high risk of metastatic disease. For unclear lesions, fluorescence in situ hybridisation (FISH), polymerase chain reaction (PCR), or conventional cytogenetics can be useful to detect t(17;22) (q22;q13), which is a distinctive feature of DFSP.

Several tumours may resemble DFSP. The most common differential diagnoses include neurofibroma, schwannoma, malignant peripheral nerve sheath tumour (MPNST), solitary fibrous tumour (SFT), leiomyosarcoma, myxoid liposarcoma, and desmoplastic melanoma. Notably, in this review, 22% of the final diagnoses were inconsistent with initial pathological diagnosis. In particular, the most common misleading diagnosis was that of dermatofibroma—the benign counterpart of DFSP—generally composed of a mixture of spindle cells and inflammatory cells, with a minor subcutaneous involvement, that could be differentiated from DFSP by negative staining for CD34. The other reported misdiagnoses were histiocytoma, fibrosarcoma, leiomyosarcoma, and neurofibroma.

The misdiagnosis is often due to inadequate tissue sampling or superficial biopsy; NCCN guidelines recommend a punch or incisional biopsy, including the deeper subcutaneous layer [42].

The interval between the clinical presentation of the lesion and first surgery can be considered prognostically relevant. In the present review, the longest was the interval between first presentation and surgery, and the largest was the tumour volume, with wider resection necessary to reach surgical free margins.

After preliminary workup, with haematoxylin and eosin (H&E) and immunopanel (i.e., for CD34 positivity), patients should be submitted for an accurate clinical exam, followed by multidisciplinary consultation and MRI with contrast, to plan appropriate treatment [42].

Wide surgical excision without lymphadenectomy is the gold standard for the treatment of this disease for both primary and recurrent lesions. To minimise the consequences of tissue defect, optimise the aesthetic result, and reduce the risk of relapse, surgery should be proposed at first appearance of the disease and performed by a surgeon with extensive expertise in vulvar surgery. Mohs micrographic surgery helped two patients in obtaining free margins and ensuring the complete resection of DFSP [43]. Excision with Mohs or other forms of margin assessment should be used; for unresectable disease, neoadjuvant Imatinib could be considered, following the execution of tumour mutation analysis.

Adjuvant treatment in cases of surgical free margins is not recommended. Radiation therapy can be advised in cases of positive surgical margins, when further resection is not feasible.

Limited long-term follow up information was reported. The prognosis in terms of disease-free survival is negatively affected by lesion size and positive surgical margins. Interestingly, recurrences were documented even after several years, suggesting a recommendation for long-term follow up. Patients should be informed about the peculiarity of the disease and educated to conduct regular self-examinations. Clinical follow-up should be integrated with MRI surveillance.

In the setting of recurrent disease, patients should be evaluated for repeated surgery or radiotherapy if resection is not feasible. When the disease is not resectable, or in the metastatic setting, treatment with Imatinib can be considered [42].

## 5. Conclusions

DFSP of the vulva is a slow-growing entity and surgery is the mainstay of treatment in this disease. Patients should be encouraged to seek medical attention when a new lesion—even apparently benign—persists or grows. A timely correct pathological diagnosis is essential to ensure proper management and limit the morbidities associated with surgical excision. Given the rarity of this disease, patients should be referred to high-volume centres to discuss diagnostic and therapeutic issues. Multicentre collaboration is essential for polling data and increasing the knowledge on this rare disease. 

## Figures and Tables

**Figure 1 cancers-16-00222-f001:**
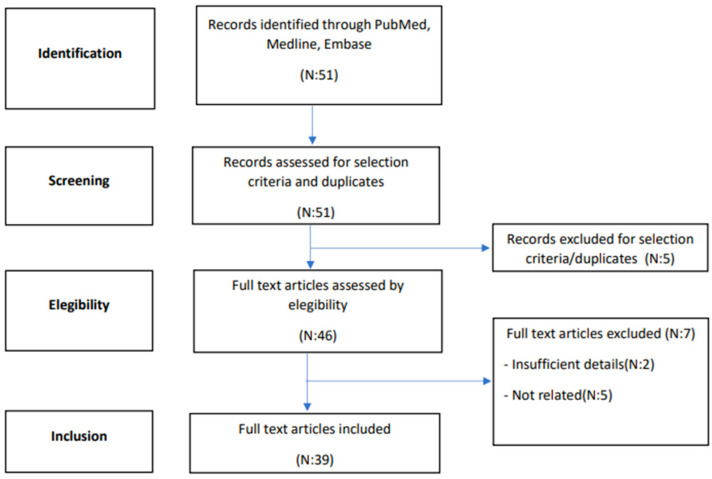
Flowchart summarising the systematic literature review process.

**Figure 2 cancers-16-00222-f002:**
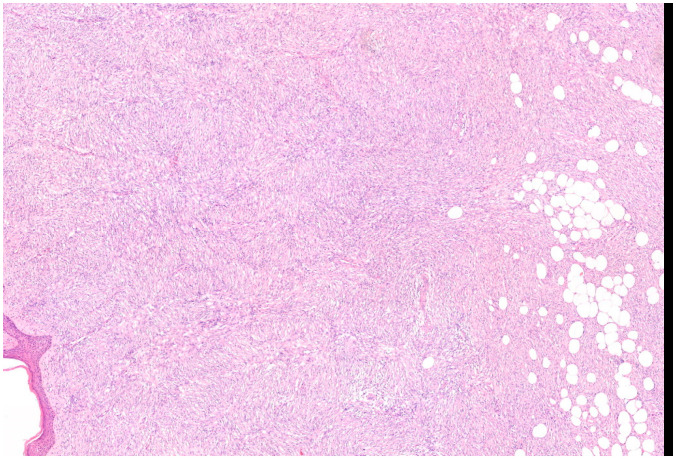
Representative case of vulvar DFSP. Spindle cell proliferation infiltrates the dermis at full thickness and permeates the subcutis, saving some lobules of adipocytes (Haematoxylin–Eosin; 50×).

**Figure 3 cancers-16-00222-f003:**
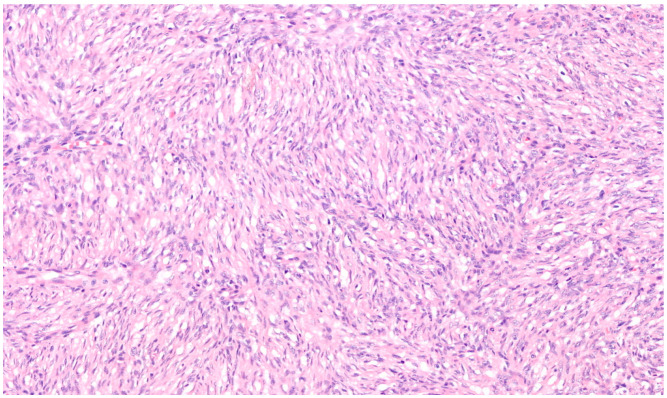
Spindle cells have elongated and mildly hyperchromatic nuclei and are arranged in a storiform pattern, generally around small vessels (Haematoxylin–Eosin; 200×).

**Table 1 cancers-16-00222-t001:** Summary of the clinicopathological characteristics of cases of vulvar DFSP reported in the literature until March 2023.

Author, Year	N	Age	Symptoms	Max Diameter (cm)	Duration (mos)	Site	CD34	COL1A1-PDGFB t 17;22 (q22;q13) Translocation	Initial Diagnosis
Aartsen, 1994 [4]	1	50	None	1.2	NA	NA	NA	NA	DFSP
Agress, 1983 [5]	1	61	None	4 8	NA	1974 right shoulder + LLM 1979 Mons pubis + contiguous vulva 1981 LLM + mons pubis	NA	NA	Histiocytoma DFSP DFSP
Alverez-Canas, 1996 [6]	1	58	None	3.2	6	LLM	+	NA	DFSP
Barnhill, 1988 [7]	1	42 45	None	1 1.5	NA	RV Paraclitoral RV Paraclitoral	NA	NA	DF DFSP
Barrios Barreto, 2022 [8]	1	54	None	12	NA	Labia majora	+	+	DFSP
Bernárdez, 2015 [9]	1	39	Pain, Malaise	12	120	LLM	+	NA	DFSP
Bertolli, 2014 [10]	2	28 57	None	5 NA	NA NA	Mons pubis RLM	NA	NA	DFSP
Bock, 1985 [11]	1	52	Vulvar swelling Itching	8	120	Mons pubis	NA	NA	DFSP
Bogani, 2014 [12]	1	48	Itching	2	24	LLM	NA	NA	DFSP
Davos and Abell, 1976 [13]	1	38	None	NA	NA	Labia majora	NA	NA	DFSP
Doufekas, 2009 [14]		39	None	NA	NA	LLM	NA	NA	DFSP
Edelweiss, 2010 [15]	13	Range 23-76	NA	Median 4 Range 1.2–15	NA	10 labia majora 2 paraclitoral mass 1 mons pubis	+11/13 −2/13	+8/11	6 DFSP 2 LG sarcoma vs. cellular NF 1 cellular DF 1 fibrosarcoma 1 LG malignant schwannoma 1 desmoplastic melanoma 1 NF vs. LG MPNST
Ghorbani, 1999 [16]	4	48 (range 44–66)	3 None 1 Pain	3–5	Mean 3 mos Range 1–6 mos	1 paraclitoral mass 2 labia majora 1 mons pubis	+4/4	NA	2 LGMPNST 1 fibrosarcoma 1 NF vs. cell leiomyoma
Gilani, 2014 [17]	1	61	None	NA	NA	mons pubis	+	NA	DFSP
Goyal, 2021 [18]	1	35	Local discomfort	6	12 + First 7 years back + R after 6 months	RLM, mons pubis	+	NA	DFSP
Hammonds, 2010 [19]	1	59	None	4	72	RV	NA	NA	DFSP
Hancox, 2008 [20]	1	55	NA	8	7	RLM	NA	NA	DFSP
Jahanseir, 2018 [21]	11	Range 29 Mean 46	NA	(Range: 2–6.3) Mean 4	NA	Vulva, Bartholin gland	+11/11	+(9/11)	2DFSP 4NA 2NF 1LGDLS 2SCN
Jeremic, 2019 [22]	1	55 (36)	None	16	18	Mons pubis, clitoris, labia majora	+	NA	DFSP
Karlen, 1996 [23]	1	36	None	5 + smaller nodules	132	Labia majora	NA	NA	DFSP
Leake, 1991 [24]	2	37 (59) 61	None	6 5 8	24 24	Labia majora + mons pubis	NA	NA	DFSP
Merlo, 2021 [25]	1	38	None	11	48	LLM + mons pubis + left thigh	+	NA	DFSP
Messalli, 2012 [26]	1	42	None	3	192	RLM	+	NA	DFSP
Moodley, 2000 [27]	1	39	None	12	Unk	LLM	+	NA	DFSP
Neff, 2019 [28]	1	57	Bleeding (Ulceration)	20	24	LLM, mons pubis	+	+	Fibrosarcomatous variant DFSP
Nirenberg, 1995 [29]	2	41 29	None None	8 3.7	Unk	Labia majora Labia majora	NA	NA	DFSP
Oge, 2009 [30]	1	56	None	3	18	Unk	+	NA	DFSP
Ohlinger, 2004 [31]	1	36	None	2.8	12	LV	+	NA	Neurofibroma
Ozmen, 2013 [32]	1	60	None	6	NA	LV–groin	+	NA	DFSP
Panidis, 1993 [33]	1	30	None	2	4	Labia majora	NA	NA	DFSP
Pascual, 2010 [34]	1	38	None	5.7	NA	Labia majora	NA	NA	DFSP
Schwartz, 1999 [35]	1	19	None	2 8	1	LLM + mons pubis	NA	NA	DFSP
Soergel, 1998 [36]	1	47	Dyspareunia, pain	3	8	LV	+	NA	DFSP
Soltan, 1981 [37]	1	83	Swelling	8.5	10	LV (Labia majora + minus)	NA	NA	DFSP
Vanni, 1999 [38]	1	39	None	6	12	Centre vulva	+	+	DFSP
Vathiotis, 2018 [39]	1	72	None	NA	48	RLM	+	+	Fibrosarcomatous DFSP
Wiszniewska, 2016 [40]	1	44	None	5 + satellite nodule 2	24	RLM	+ Myxoid areas CD34-	+	Neurofibroma
Zemni, 2019 [41]	1	47	Pain	6	4	Labia majora + groin	+	NA	DFSP
Zlatnik, 1999 [42]	2	61 34	None None	5 + satellite nodules (2) Unk	60 252	Mons pubis LLM	NA NA	NA NA	DFSP DFSP
Mancari R, 2024	1	64	None	4	12	Labia majora	+	NA	DFSP

Table legend: Unk: unknown; NA: not available; LG: low grade; RV: right vulva; LV: left vulva; RLM: right labium majus; LLM: left labium majus; DFSP: dermatofibrosarcoma protuberans; DF: dermatofibroma; NF: neurofibroma; LGMPNST: low-grade peripheral nerve sheet tumour; LGDLS: low-grade dedifferentiated liposarcoma; SCN: spindle cell neoplasm.

**Table 2 cancers-16-00222-t002:** Summary of the treatment approaches and disease course of vulvar DFSP cases reported in the literature until March 2023.

Author, Year	N	Initial Treatment	Free Margins	Repeated Surgery for Positive Margins (Type)	Adjuvant Therapy	Recurrence (Site)	DFS (mos)	Treatment of Recurrence
Aartsen, 1994 [4]	1	LE	N	Y (WLE, Radical vulvectomy)	N	N	72 NED	
Agress, 1983 [5]	1	LE	Unk	N	N	Y (2 local)	60 5	WLE Radical vulvectomy
Alverez-Canas, 1996 [6]	1	LE	N	Y (WLE)	N	N	11 NED	
Barnhill, 1988 [7]	1	LE	Unk	N	N	Y (local)	36	WLE + Radical vulvectomy
Barrios Barreto, 2022 [8]	1	WLE	Y	N	N	Y (local)	6	NACHT (Imatinib), then WLE
Bernárdez, 2015 [9]	1	Radical excision	N	N	RT	N	12 NED	
Bertolli, 2014 [10]	2	Vulvectomy LE	Y N	N Y (2 × WLE)	N N	N N	40 NED 10 NED	
Bock, 1985 [11]	1	WLE	Y	N	N	N	6 NED	
Bogani, 2014 [12]	1	WLE	Y	Y (for close margins)	N	N	24	
Davos & Abell, 1976 [13]	1	WLE	Y	N	N	N	240 NED	
Doufekas, 2009 [14]	1	WLE	N	Y (MMS)	N	N	36 NED	
Edelweiss 2010 [15]	13	3LE 7 LE+WLE 2 WLE 1 LE+ radical vulvectomy	6 N 6 Y 1 Unk	7 WLE 1 radical vulvectomy	N	Y 7 local (5 with positive margins 1 with margins NA 1 with negative margins	27 48 7	3WLE 1 WLE + partial vulvectomy 1 incomplete LE + Imatinib + 1 WLE + RT + CHT
Ghorbani, 1999 [16]	4	4 LE	1 Unk 3 N	N 2Y (WLE) 1 refused surgery	N 3N	Y (10 times, DFS 2 year, hemi-vulvectomy) 2 N AWD	24 6 144	WLE, NED 7 years after last surgery
Gilani, 2014 [17]	1	LE	N	Y (2 × WLE)	N	N	12 NED	
Goyal, 2021 [18]	1	WLE	Unk	Y	N	Y (2 local)	6 30	WLE Radical hemi-vulvectomy
Hammonds, 2010 [19]	1	MMS	Y	N	N	N	30 NED	
Hancox 2008 [20]	1	LE	Y (2LE + MMS)	Unk	N	N	129 NED	
Jahanseir, 2018 [21]	11	9 LE 2 WLE	Y (8) Y (2)	3Y 1Y	1 RT → NED 35	Y (local)	6	
Jeremic, 2019 [22]	1	Radical vulvectomy	Y	N	N	Y (lung mets)	18 DOD	
Karlen, 1996 [23]	1	WLE	Y	N	N	N	27 NED	
Leake, 1991 [24]	2	LE LE	N N	Y, WLE N	N N	N Y (local)	18 NED 24	Radical hemi-vulvectomy
Merlo, 2021 [25]	1	NACHT (Imatinib, PR) → Radical 1 hemi-vulvectomy + inguinal LND + WLE left thigh	Y	N	N	N	15 NED	
Messalli, 2012 [26]	1	WLE	N	Y (WLE)	N			
Moodley, 2000 [27]	1	WLE	Y	Y (WLE)	N	N	3 NED	
Neff, 2019 [28]	1	Radical vulvectomy + LND	Y	N	Imatinib for 12 months	N	18 NED	
Nirenberg, 1995 [29]	2	WLE WLE	Y Y	N N	N N	N N	30 17	
Oge, 2009 [30]	1	LE	N	Y (2 × WLE)	N	N	15 NED	
Ohlinger, 2004 [31]	1	LE	Y	N	N	Y (4 local)	12	5 × WLE
Ozmen, 2013 [32]	1	WLE	Y	N	N	N	24 NED	
Panidis, 1993 [33]	1	LE	N	N	N	Y (local)	6	Radical vulvectomy
Pascual, 2010 [34]	1	LE	N	Y (MMS)	N	N	15 NED	
Schwartz, 1999 [35]	1	WLE	Y	N	N	Y (local)	2	WLE + inguinal LND (LN neg)
Soergel, 1998 [36]	1	LE	N	Y (partial radical vulvectomy)	N	Y (local) Y distant (abdomen + lung)	16 5	WLE + RT Excision of abdominal mass + CT (PD) DOD
Soltan, 1981 [37]	1	WLE	Y	N	N	N	6 NED	
Vanni, 1999 [38]	1	WLE	Y	N	N	N	24 NED	
Vathiotis, 2018 [39]	1	LE	N	N	N	Y (2 local, 1 distant) 1 local 2 local 3 lung met	18 5 23	WLE+ Radical vulvectomy Imatinib
Wiszniewska, 2016 [40]	1	BPS	N	Y (WLE)	N	N	18 NED	
Zemni, 2019 [41]	1	WLE	Y	N	N	N	1 NED	
Zlatnik, 1999 [42]	2	WLE + inguinal LND LE	N N	Y (WLE) Y (left hemi-vulvectomy + inguinal LND—WLE)	N N	N N	108 NED 96 NED	
Mancari, 2024	1	WLE	N	Y (WLE)	N	N	NED	

Table legend: Y: Yes; N: No; Unk: unknown; BPS: biopsy; LE: local excision; WLE: wide local excision; LN: lymph nodes; LND: lymph node dissection; MMS: Mohs micrographic surgery, RT: radiotherapy; NACHT: neoadjuvant chemotherapy; CHT: chemotherapy; NED: no evidence of disease; AWD alive with disease; PD: progression disease; DOD: dead of disease.

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
