# Peer review of "Dermatofibrosarcoma Protuberans of the Vulva: A Review of the MITO Rare Cancer Group"

_cancers, 2024, doi:10.3390/cancers16010222_

Round 1
Reviewer 1 Report
Comments and Suggestions for Authors
Thank you for asking me to review this interesting article. It is a useful reference/summary of what is known about this rare condition for all working in this field. I recommend it for publication pending some editing of English, as in parts it was difficult to understand due to the translation from Italian.
Comments on the Quality of English LanguageAs above - moderate editing of English required.
Reviewer 2 Report
Comments and Suggestions for Authors
Thank for the opportunity to review the interesting paper entitled "Dermatofibrosarcoma Protuberans of The Vulva: A Review of The MITO Rare Cancer Group". The Authors reported a systematic review of the literature about the vulvar dermatofibrosarcoma protuberans (DFSP). In general, this a well written report focusing on the management of a rare tumor as DFSP which merits publications. The introduction adequately describes the background and highlights the rarity of the disease and the absence of management guidelines. The article emphasizes the importance of a multidisciplinary approach due to the disease's rarity, challenges in diagnosis, and the need for early detection to facilitate conservative surgery. Due to lack of guidelines that indicates the management, this review of the current literature can be helpful for daily clinical practice.
Please correct:
Line 99 "The mean time from mass appearance to treatment in the included cases was 52.6 99 months (range 1-252). Please add to the mean the standard deviation or median time plus range.
Comments on the Quality of English Language
Good quality of English
Reviewer 3 Report
Comments and Suggestions for Authors
The authors presented this review entitled “Dermatofibrosarcoma Protuberans of the vulva: a Review of the MITO Rare Cancer Group”. Some references are not updated. Treatment strategies, such as European consensus or NCCN guidelines for dermatofibrosarcoma protuberans - update, are not mentioned thoroughly.
Comments on the Quality of English LanguageMinor editing of English language required
Reviewer 4 Report
Comments and Suggestions for Authors
Excellent article, proper systematic review and interpretation of results. I have one question if it is feasible to perform a survival analysis with Kaplan mayer curves?
The main question in the manuscript is the management and the ways to cure the local recurrence. The topic is original, systematic reviews add in a scientific way the potential management options of the disease Systematic review offer greater level of evidence, they offer data about the mean age of patients, localization and stage of disease also options of surgical management and adjuvant therapy The manuscript would merit of a metanalysis of clinical data obtained by the studies.The conclusions are consistent with the evidence and arguments presented
and they address the main question posed. The references are appropriate. There are no additional comments on the tables and figures.Author Response
Please see the attachment.

Reviewer 5 Report
Comments and Suggestions for Authors
Well described review concerning the DFSP tumors. Some photos of the lesion would be helpful.
-------------------------------------------------------------------------------------
The paper is a systematic review devoted to the rare tumor presented on the vulva.The tumor is so rare and extremely seldom recognized that the topic is original, however, due to the rarity of this pathology I am not convinced that it is relevant in the field.
The paper summarizes data concerning DFSP tumors presented so far in the Pubmed and Medline databases, and basing on them presents the diagnosis and treatment as well as prognosis of that rare tumors
No remarks regarding the methodology. I would advise to present some photos of the vulvar tumors.
The conclusions are consistent with the evidence and arguments presented
and they address the main question posed.
The references are appropriate.
There are no additional comments regarding the tables and figures.
